Evaluation of the effects of the green nanoparticles zinc oxide on monosodium glutamate-induced toxicity in the brain of rats

Hamza Reham Z. reham.z@tu.edu.sa reham_z@Zu.edu.eg 1 2
Al-Salmi Fawziah A. 2
El-Shenawy Nahla S. 3
1 Department of Zoology, Faculty of Science, Zagazig University , Zagazig , Egypt
2 Department of Biology, Faculty of Science, Taif University , Taif , Saudi Arabia
3 Department of Zoology, Faculty of Science, Suez Canal University , Ismailia , Egypt
Anderson Todd
Electronic publication date: 2019 Sep 23
Publication date: 2019
Volume: 7
Electronic Location ID: e7460
Received 2019 May 18; Accepted 2019 Jul 11
Copyright: ©2019 Hamza et al.
Copyright year: 2019
Copyright holder: Hamza et al.
License: This is an open access article distributed under the terms of the Creative Commons Attribution License, which permits unrestricted use, distribution, reproduction and adaptation in any medium and for any purpose provided that it is properly attributed. For attribution, the original author(s), title, publication source (PeerJ) and either DOI or URL of the article must be cited.
License URL: https://creativecommons.org/licenses/by/4.0/

Keywords: Zinc oxide nanoparticles; Green synthesis, Neurotransmitters, MSG, Cyclooxygenase-2, Brain, Brain biomarkers, Acetylcholinesterase, Histopathology

Funding: The authors received no funding for this work.

==============================
Background

Monosodium glutamate (MSG) is used extensively as a food additive in the diets of many countries around the world.

Aim of the study

Our aim was to determine the effects of green zinc oxide nanoparticles on MSG-induced oxidative damage, neurotransmitter changes, and histopathological alternation in the cerebral cortexes of rats.

Methods

MSG was administered orally at two doses of 6 and 17.5 mg/kg body weight. The higher dose was associated with a significant decline in the activities of superoxide dismutase, catalase, and glutathione peroxidase, as well as the levels of brain-derived neurotrophic factor (BDNF) and glutathione (GSH) in the cerebral cortex of rats.

Results

The administration of zinc oxide nanoparticles/green tea extract (ZnO NPs/GTE) to 17.5 mg/kg MSG-treated rats was associated with significant improvements in all parameters previously shown to be altered by MSG. The higher dose of MSG induced significant histopathological variation in brain tissue. Co-treatment of rats with ZnO NPs/GTE and MSG-HD inhibited the reduction of neurotransmitters and acetylcholinesterase by MSG.

Conclusions

ZnO NPs/GTE have the potential to protect against oxidative stress and neuronal necrosis induced by MSG-HD. ZnO NPs/GTE conferred a greater benefit than the control treatment or ZnO NPs or GTE administered separately.

Introduction

Metal oxide nanostructures are one of the most extensively fabricated types of nanomaterials (Chen et al., 2017). There are many forms of mineral metal oxides have been synthesized and continue to exist as TiO2, CuO, and ZnO. Zinc oxide nanoparticles (ZnO NPs) are of maximum attentiveness because they are low-cost to produce, safety makes ZnO a good candidate for use in food preservation and can be formed efficiently. ZnO NPs have extensive application in artificial flavoring (Catherine, Adam & Curtis, 2003), cosmetics, sunscreens, and batteries (Ma, Williams & Diamond, 2013). Also, the biological applications of ZnO NPs have a great role in the biosensor and medical devices. ZnO NPS has anticancer effects (Wahab et al., 2013), antibacterial activity (Dwivedi et al., 2014), agriculture, and biomedicine applications (Sirelkhatim et al., 2015). There was a concept that ZnO NPs can be applied to some biomedical materials (Berube, 2008).

However, Afifi, Almaghrabi & Kadasa1 (2015) found that ZnO NPs had a restorative effect on male reproducing functions. It had adverse effects on animals and human health because the size of the NPs surface area greatly augments their capability to produce ROS through oxidative stress (Moller et al., 2010). It has the excessive potential to pass through the placenta and testis barriers to efficient targeting of cells and causing many diseases (Al-Salmi, Hamza & El-Shenawy, 2019). Therefore, it is obligatory to evaluate its toxicity of ZnO NPs in distinctive organs for long- and the short-period of study. Recently, the effects of ZnO NPs on the liver and renal tissues had been evaluated in the laboratory (Al-Salmi, Hamza & El-Shenawy, 2019; Zanchi et al., 2015; El-Shenawy et al., in press).

Moreover, biosynthesis of nanoparticles using microorganisms and plants had biomedical applications (Agarwal, Kumar & Rajeshkumar, 2017). They allow the synthesis of ZnO NPs with the great purities and show more activity. Using extracts of any parts of the plant is a very safe to the ecological environment, cheap in its preparation and it does not use any transitional groups. It takes very short-time and produces a pure product free of any contaminants (Heinlaan et al., 2008). For example, spherical shaped ZnO NPs were created by Aloe Vera leaf extract where the active groups of plant extract acted as both reducing and eclipse agent. Santhoshkumar, Kumar & Rajeshkumar (2017) found that ZnO NPs using plant leaf extract (P. caerulea L.) can be worked against the urinary tract infections (Rajakumar et al., 2018).

In spite of the fact that the NPs biosynthesis by different plant extracts is vague, it has been showed that the biomolecules in plant extract (protein, phenol, and flavonoids) play a noteworthy part within the diminishment of metals particles and eclipsing the biosynthesized nanoparticles (Krishnaraj et al., 2010).

Green tea (Camellia sinensis) is one of the most popular drinks around the world (Figueiroa et al., 2009). It is thought to have profitable effects on wellbeing due to its high amount of polyphenols as epigallocatechin-3-gallate (EGCG), which has anti-oxidative properties (Soussi et al. (2006)) and repress reactive oxygen species (ROS) activity in a dose-dependent way. Polyphenols from the green tea appear to have 20 times more impressive antioxidant activity than vitamin C (Rajakumar et al., 2018; Elhalwagy, Darwish & Zaher, 2008). Abshenas et al. (2011) reported that green tea extract could reduce the conflicting effects of hyperthermia on semen parameters in a short duration.

Based on previous studies, MSG has a toxic effect on the brain by increasing the level of neurotransmitters of the nervous system of mammals which play an essential role in both physiological and pathological processes (Mattson, 2008). Also, it increased in intracellular calcium which in turn promotes an arrangement of enzymatic response which leads to cell death. Besides, it has been shown that MSG utilization causes obsessive changes in brain structure related with neuronal damage and articulated oxidation stress (Onaolapo et al., 2016). MSG overdose has been shown to induce neurodegenerative injuries, Parkinson’s disease, and epilepsy (Narayanan et al., 2010). It is known to stimulate the discharge of norepinephrine (NE) and dopamine (DA) from a number of different brain regions (Jhamandas & Marien, 1987).

Serotonin, or 5 hydroxytryptamine (5HT), could be a biogenic amine affecting numerous functions of the central nervous system. It is stored in vesicles and released into the blood plasma as a response to certain stimuli (Bertrand & Bertrand, 2010). An expanded brain serotonin level is recommended to enhance the cognitive performance. However, a decline in brain 5HT and its precursor tryptophan contents has been shown to destroy memory ability (Laercio et al., 2004).

It has been entrenched that a number of bioactive markers participate in neuropathic pain, such as cyclooxygenase −2 (COX-2), and prostaglandin E2 (PGE2) (Li, Van Aelst & Cline, 2000). BDNF is the richest neurotrophin in the brain and fundamental for neuronal survival during the development of brain’s neurons (Waterhouse & Xu, 2009). BDNF is also disclosed in the liver (Cassiman et al., 2001), skeletal muscle (Matthews et al., 2009) and cardiovascular system (Donovan et al., 2000). Cyclooxygenase (COX) is responsible for prostaglandin (PG) H formation by synthesis of prostanoids of arachidonic acid metabolites forming PGs, prostacyclin, and thromboxanes (Minghetti, 2004). COX-2 is asserted in many cell types as the effect of growth factors, cytokines, and pro-inflammatory variables.

The role of ZnO NPs /GTE against MSG toxicity on the brain has not studied until now, except on the liver. In the ongoing study, we studied the neuroprotective effect of ZnO NPs /GTE against MSG neurotoxicity. The effect of ZnO NPs/GTE on the brains of rats was evaluated on the bases of assessment of some neurotransmitters, redox status assays, and histopathology changes.

Experimental procedure

Zinc oxide nanoparticles (ZnO NPs) properties and chemicals

The ZnO nanoparticles are a white powder with 200 nm diameter, 20–40 nm size, and purity 99.5% (Sigma-Aldrich Company). The morphology and particle size of ZnO NPs were characterized using scanning electron microscopy (SEM, JSM-6301, Japan) (Al-Salmi, Hamza & El-Shenawy, 2019) and Energy Dispersive X-ray Diffractive, EDX, respectively.

Preparation of green tea extract, MSG and green ZnO NP complex

The green tea extract (GTE) was processed as described before by Zanchi et al. (2015). The stock solution of MSG was provided by dissolving 60 g in 1,000 mL of distilled water. The process of the GTE conjugation with ZnO NPs was explained previously in our laboratory (Al-Salmi, Hamza & El-Shenawy, 2019).

Animals and ethical consideration

In this study, adult male rats weighed 200–250 g were used. They were purchased from the Faculty of Pharmacy (Zagazig University, Egypt). They were used after two weeks of the acclimatization. The animals were housed beneath the ordinary circumstances of temperature (23 °C) and were given drinking tap water and rodent nourishment pellets. The rats were chosen as the foremost appropriate animal species for the experiment as they are small, easily housed and maintained, and adapt well to new surroundings, easy to handle, they are mimic human greatly , so we can test these parameters effectively.

The sample size of rats was chosen based on the less number of animals to get actual results.

This experimental study was performed with the confirmation of the local ethics committee on use and care for animal experiments at Taif University of Biological Department (permit number: 39-31-0034).

Experimental protocol

There were eight groups (n = 8 each) that were treated as the following; vehicle (for control), ZnO NPs (10 mg/kg), GTE (250 mg/Kg), and GTE/ZnO NPs complex as mentioned in groups two and three. The other four groups were treated with the lower dose of MSG (6 mg/Kg), the higher dose of MSG (17.5 mg/Kg), MSG-LD + GTE/ZnO NPs complex and MSG-HD + GTE/ZnO NPs complex. All the different treatment was given to the rats orally by feeding needle. The rats were killed by sudden decapitation and the brains were removed for more evaluation after 30 days of treatment.

Acetylcholinesterase estimation

The blood samples were collected from the retro-orbital plexus vein and were centrifuged at 3,000× g for 15 min. Serum was collected and frozen at −20 °C until the biochemical estimation. Acetylcholinesterase (AChE) in the serum was estimated by the kit of Bio-diagnostic Company, Dokki, Giza, Egypt, according to Minghetti (2004).

Brain preparation and estimation of neurotransmitters

Rats were killed by sudden beheading after 30 days of treatment. The brain was quickly and carefully dissected on dry ice glass plate between 8.00 and 9.00 a.m for all the groups and was reserved in a freezer at −20 °C. The brain tissues were cut coronal to get the cerebral cortex. The brain samples were split into two parcels: the primary one was utilized for the planning of brain homogenates and the second was urilized for histological examination.

The first set of the cerebral cortex was weighed and homogenized in a cold acidified n-butanol to obtain 10% homogenate. The internal standard of serotonin, norepinephrine, and dopamine was prepared by adding 0.3 mL standard mixture (100 µL containing 100 µg of each neurotransmitter) to 9.7 mL of 0.2N acetic acid. Aliquots of 200 µL of this solution were diluted to 300 µL with 0.2N acetic acid then, three mL of acidified n-butanol was added (0.85 mL of concentrated HCl/L n-butanol) and internal standard tubes were centrifuged at 10,000× g for 5 min. The aqueous phase was collected and divided into two portions; the first one was used for serotonin determination while the other was used for epinephrine, norepinephrine, and dopamine assessment.

Serotonin content was carried out depending on the reaction of ortho-phthalaldehyde (Magnotti, Eberly & McConnell, 1988) and according to the instructions of the manufacturer of ELISA kits. The epinephrine, norepinephrine, and dopamine estimated by the method of Ciarlone (Magnotti, Eberly & McConnell, 1988) that depends on the oxidation by iodine that consists of acetic acid (5 N), alcoholic iodine solution (0.1 N), disodium ethylene diamine tetra-acetic acid dehydrate (0.1 N) in sodium acetate (pH 6.7-7) and alkaline sulfite (5 N) that prepared just before use.

Determination of redox status of the brain

The superoxide dismutase (SOD) activity was evaluated using the assay kit and was presented as U/g. Catalase (CAT) activity was measured according to an assay kit and was expressed as mmol/g tissue. All the kits brought from Bio-Diagnostic Company CO, Giza, Egypt.

Glutathione (GSH) level and glutathione peroxidase (GPx) activity were determined using the method of the commercial kit from Bio-Diagnostic.

Myeloperoxidase (MPO) activity was measured by Ciarlone (1978) and Suzuki et al. (1983). Xanthine oxidase (XO) activity was assayed spectrophotometrically according to Litwack et al. (1953).

Total thiols level was dogged by the method of Hu (1994) and was presented as mmol/g tissue.

Estimation of some bioactive compounds markers for neuroinflammation

The cerebral cortex levels of BDNF, COX-2 and PGE2 in supernatant (cerebral cortex was homogenized in two fold volumes of 0.01 mol/L phosphate-buffered saline that containing 0.05% Tween-80 and centrifugation at 15,000 g at 4 °C for 20 min) were measured by (ELISA) with monoclonal antibodies particular for rats BDNF, COX-2, and PGE2. The commercial ELISA kit of rat BDNF was acquired from WKEA Co. The ELISA kit of rat COX-2 and PGE2 were purchased from Immuno-Biological Laboratories, Inc., Minneapolis, MN, USA.

Sample preparation for light microscopy and histopathological analysis

At the end of the experimental time, the second set of the cerebral cortex was fixed in 10% formal saline, dehydrated in a graded series of ethanol and finally, embedded in paraffin. The samples were sectioned by microtome into 3–4 m thickness and stained with hematoxylin and eosin (H&E) for histopathological examinations employig a light microscope.

Statistical analysis

The data were analyzed by using the SPSS software, version 20. Statistical significance between groups was computed by investigation of variance and analyzed by one-way ANOVA and Tukey’s test. The contrasts were plan to be significant at P ≤ 0.05.

Results

Structural and morphological characterization

The SEM image of the sample is shown in Fig. 1A. The particles can be observed as spherical or quasi-spherical particles are dominant. The chemical compositions of the zinc oxide/green tea complex were determined using EDX (Fig. 1B). The study was carried out for the synthesized ZnO NPs complex with GTE to know the complex elemental composition. The peaks of the two essential elements like carbon and oxygen, and respective Zn(II) elements, which constitute the molecules of ZnO NPs/GTE complex is clearly identified as shown in Fig. 1B and this analysis showed the peaks that corresponded to the optical absorption of the ZnO NPs/GTE complex. The elemental analysis of the complex yielded 40.77% of zinc, 30.18% of oxygen and 29.05% of carbon which proves that the produced ZnO NPs complex with GTE is in its highest purified formula and confirmed it’s a combination.

Figure 1 Scanning Electron Microscopy (SEM) analysis showing the morphology of the prepared ZnO nanoparticles.

(A and B) showing two views of the surface morphology of the ZnO nanoparticles complexes with green tea. The SEM image showed that most of the ZnO nanoparticles showed a homogeneous distribution in shape. (C) Energy Dispersive X-Ray Diffractive (EDX) Analysis showing the chemical compositions of the zinc oxide/green tea complex was determined using energy-dispersive X-ray diffraction (EDX).

Table 1 Effect of ZnO NPs/GTE mixture on the dopamine, serotonin, catecholamine, acetylcholinesterase and thiol of brain’s tissue of rats previously exposed to MSG.

Parameters	Control	MSG-LD	MSG-HD	ZnO NPs	GTE	ZnO NPs/GTE	MSG-LD + ZnO NPs/GTE	MSG-HD + ZnO NPs/GTE	
Dopamine (DA)(/mL)	1.6 ± 0.2	1.7 ± 0.2	0.8 ± 0.1a	1.5 ± 0.1	1.6 ± 0.1	1.9 ± 0.1	1.3 ± 0.1	1.1 ±0.1b	
Serotonin (5-HT) (µg/ mL)	0.7 ± 0.1	0.8 ± 0.1	0.4 ± 0.1a	0.6 ± 0.1	0.6 ± 0.1	0.7 ± 0.1	0.6 ± 0.1	0.6 ± 0.1	
Epinephrine (µg/ mL)	0.9 ± 0.1	1.0 ± 0.1	0.6 ± 0.1a	0.8 ± 0.1	0.8 ± 0.1	0.9 ± 0.1	0.8 ± 0.1	0.8 ± 0.1b	
Norepinephrine (µg/mL)	0.8 ± 0.1	0.8 ± 0.1	0.4 ± 0.1a	0.7 ± 0.1	0.8 ± 0.1	0.8 ± 0.1	0.7 ± 0.1	0.6 ± 0.1b	
Acetylcholinesterase (mmol/min/mL)	592.1 ± 22.9	607.8 ± 14.1	382.2 ± 27.2a	564.4 ± 17.3	564.7 ± 15.8	626.4 ± 10.1	573.4 ± 22.2b	467.1 ± 11.6b	
Thiol level	11.5 ± 0.9	11.8 ± 1.4	6.2 ± 1.4a	13.8 ± 0.8a	11.6 ± 0.8	11.1 ± 1.0	12.2 ± 1.6	10.1 ± 1.7b	
Notes.

Values are expressed as mean ± SE, n = 8.

MSG- LD the lower dose of monosodium glutamate

MSG-HD the higher dose of monosodium glutamate

ZnO NPs zinc oxide nanoparticles

GTE green tea extract

a Significant difference as compared to control.

b significant difference as compared to its relative group of MSG (P ≤ 0.05).

Hormones, AchE and thiol levels

Effect of ZnO NPs/GTE mixture on the dopamine, serotonin, catecholamine, acetylcholinesterase, and thiol were presented in Table 1. Significant declined in catecholamine level (noradrenaline and adrenaline), dopamine, and serotonin was detected in the higher dose of MSG in comparing with normal animals. In addition to neurotransmitters, significant differences were also detected in the activity of AchE and thiol levels at the 17.5 mg/kg MSG as compared with the control animals (Table 1). There were no differences between ZnO NPs or GTE groups and control rats. In the case of the treatment for the animals with ZnO NPs/GTE mixture and MSG, the values of all the previous parameters returned approximately to the control group.

Antioxidant and oxidative stress enzymatic/non-enzymatic

The SOD activity was significantly decreased in MSG-HD and ZnO NPs groups (Fig. 2). However, CAT activity decrease only in MSG-HD treated animals and did not change in all other groups (Fig. 2).

Figure 2 Activities of superoxide dismutase and catalase in the rat brains treated with different doses of monosodium glutamate (MSG), zinc oxide nanoparticles (ZnO NPs), green tea extract (GTE), and ZnO NPs/GTE.

Values are presented as the mean ± S.E. (A) P < 0.05 vs. control and (B) P < 0.05 vs. its relative group of MSG.

Figure 3 Glutathione peroxidase (GPx) activity and glutathione (GSH) levels in rat brains.

Glutathione peroxidase (GPx) activity and glutathione (GSH) levels in the rat brains treated with different doses of monosodium glutamate (MSG), zinc oxide nanoparticles (ZnO NPs), green tea extract (GTE), and ZnO NPs / GTE. Values are presented as the mean ± S.E. a P < 0.05 vs. control and b P < 0.05 vs. its relative group of MSG.

Treatment the rats with MSG at different doses (6 and 17.5 mg/kg) caused a significant decrease in GPx activity and GSH level at the higher dose only (Fig. 3). ZnO NPs only or with MSG at different doses significantly decreased the GSH and GPx of the brain rats.

No critical changes were noticed in MPO of all treated group except the MSG-HD with ZnO NPs/GTE animals (Fig. 4). XO activities were elevated in ZnO NPs as compared to control group by 1.9-fold, as well as in MSG-LD or MSG-HD with ZnO NPs/GTE as compared to its relative group of MSG by 1.5- and 1.7-fold, respectively (Fig. 3).

Figure 4 Activities of myeloperoxidase and xanthine oxidase in the rat brains.

Activities of myeloperoxidase and xanthine oxidase in the rat brains treated with different doses of monosodium glutamate (MSG), zinc oxide nanoparticles (ZnO NPs), green tea extract (GTE), and ZnO NPs/GTE. Values are presented as the mean ± S.E. (A) P < 0.05 vs. control and (B) P < 0.05 vs. its relative group of MSG.

The level of BDNF was decreased significantly in ZnO NPs- treated animals by 41.5% as compared to normal rats (Fig. 5). In both treatment doses of MSG with ZnO NPs/GTE, the BDNF was significantly decreased at the lower dose and increased at the higher dose.

Figure 5 Brain derived neurotrophic factor (BDNF) in the rat brains.

Brain derived neurotrophic factor (BDNF) in the rat brains treated with different doses of monosodium glutamate (MSG), zinc oxide nanoparticles (ZnO NPs), green tea extract (GTE), and ZnO NPs/GTE. Values are presented as the mean ± S.E. (A) P < 0.05 vs. control and (B) P < 0.05 vs. its relative group of MSG.

In MSG-HD group, the COX-2 activity elevated by 2.1- fold as compared to control (Fig. 6). In the ZnO NPs group, the COX-2 activity decreased by 33% as compared to normal animals. The combination of MSG-HD with ZnO NPs /GTE caused decreased in COX-2 by 40% as compared to MSG-HD-treated rats.

Figure 6 Cyclooxygenase-2 (COX-2) activity in the rat brains.

Cyclooxygenase-2 (COX-2) activity in the rat brains treated with different doses of monosodium glutamate (MSG), zinc oxide nanoparticles (ZnO NPs), green tea extract (GTE), and ZnO NPs/GTE. Values are presented as the mean ± S.E. (A) P < 0.05 vs. control and (B) P < 0.05 vs. its relative group of MSG.

The PGE2 level was increased by 1.8-fold in MSG-HD group as compared to control animals (Fig. 7). The MSG-LD with ZnO NPs/GTE increased the PGE2 by 1.3-fold as compared to MSG-LD group. However, The MSG-HD with ZnO NPs/GTE decreased PGE2 by 24% as compared to MSG-HD animals.

Figure 7 Prostaglandin E2 (PGE2) level in the rat brains.

Prostaglandin E2 (PGE2) level in the rat brains treated with different doses of monosodium glutamate (MSG), zinc oxide nanoparticles (ZnO NPs), green tea extract (GTE), and ZnO NPs/GTE. Values are presented as the mean ± S.E. (A) P < 0.05 vs. control and (B) P < 0.05 vs. its relative group of MSG.

Histological changes

Histopathological assessment of brain tissues revealed that no histopathological alternations were observed in control or ZnO NPs or GTE groups. All these groups of rat showed the normal arrangement of layers of neurons and neuronal fibers (Figs. 8A–8C) and the normal cerebral cortex shaped of circular and pyramidal neurons encompassed by eosinophilic glial filament were also watched.

Figure 8 Histopathological sections.

: Histopathological slides of the brain stained with hematoxylin and eosin in (A) control groups of rat showing normal arrangement of layers of neurons (Blue arrow) and neuronal fibers (*) (400X); (B) cross section of rat brain treated with ZnO NPs showing normal structure (100X). (C) the cross section showed the animals that treated with green tea extract with normal nerve cell (400X); (D) cross section of rat brain treated with ZnONPs/green tea (10 mg/Kg) showing normal white matter with normal neurons (Blue arrow) and neuronal fibers (*) (400X); (E) (MSG-LD) treated-group (75 mg/kg) showing moderate area of necrosis (***) in the brain (400X); (F) (MSG-HD) treated-group (150 mg/kg) showing disorganized pyramidal cells (*) and some swelling in the brain cells (***) (400X); (G) The MSG-LD and ZnONPs/green tea complex showing very reduced congested area (*) with normal neural fibers (Yellow arrow ) (100X); (H) the MSG-HD and ZnONPs/green tea complex showing the recovery of the congested area with scattered few apoptotic cells (**) with appearance of dark stained nuclei (**) and reduced necrotic neurons (100X).

The cross-section of the rat brain treated with ZnONPs/GTE (10 mg/Kg) showed a normal white matter with normal neurons and neuronal fibers (Fig. 8D). The moderate area of hemorrhage and necrosis in the brains of rats treated with MSG-LD (Fig. 8E). However, the brain tissue of the rats treated with MSG-HD showed a large area of hemorrhage and necrotic areas of the brain with the congested area; in addition, degeneration in some glial cells was noticed (Fig. 8F).

The lower dose of MSG and Zn ONPs/GTE reported having a very reduced the congested area with normal neural fibers (Fig. 8G). The highest dose of MSG and ZnONPs/GTE showed the recovery of the congested area with the moderate area of necrosis with normal appearance to fibers and reduced necrotic neurons (Fig. 8H).

Discussion

In the present study, the brain has been selected because it is more insightful to the oxidative stress as it contains the low concentration of antioxidants, and it needs high energy and also contains the high cellular levels of lipids and proteins. The microglial actuation produces high contents of free radicals that can harm the proteins, lipids, as well as the nucleic acids at the particle deposition location (Veronesi et al., 2005).

MSG-induced oxidative stress pathways in cerebral cortex could be by decreasing the neurotransmitters as shown in the present investigation where the levels of dopamine, serotonin, epinephrine, and norepinephrine were reduced with 17.5 mg/kg of MSG. The prolonged use of MSG-induced histopathological changes in the brain depending on the dose and the ROS formation which confirmed by declining the antioxidant enzymes activities of SOD, CAT, GPx, and reduces the level of GSH as well as the BDNF. However, the levels of COX-2 and PGE2 were significantly elevated in the cerebral cortex as the MSG-treatment.

The present study committed by other previous investigations, long-term treatment of MSG is correlated with several reversals in the nervous system (Onyema et al., 2006; Zanfirescu et al., 2018), liver and kidneys (Zanchi et al., 2015; Hamza & Al-Harbi, 2014) due to an elevation of ROS production (Gebicki, 2016).

Also, Umukoro et al. (2015) demonstrated that orally administration of 500 mg/kg of MSG to mice elevated the malondialdehyde and diminished the GSH levels in brain tissue. Similar diversity within the antioxidant enzymes activities combining ingestion of MSG was reported already in the heart tissue of rats by Singh and Ahluwalia (Singh & Ahluwalia, 2003) and liver tissue by Al-Salmi, Hamza & El-Shenawy (2019) as well as in the kidney tissue (Paul et al., 2012; Shivasharan et al., 2013). This is often an sign that the noxious impact of MSG extends to all tissues within the body. Free radicals assembled within the brain tissue due to ingestion of MSG are known to cause disintegration of most proteins including enzymes (Gebicki, 2016). The defect in blood–brain barriers and compete in the deleterious action on the brain antioxidant enzymes and other biomarkers occurred as the free radicals formation in situ and also due to its diffusion from circulation.

Dopamine which is a neurotransmitter that occur abundantly in the central nervous system (Mishra & Goel, 2013). Catecholamine is of core importance getting learning skills and also in the building the memory (Izquierdo & Medina, 1997). Moreover, alterations of serotonin levels may be correlated with defect the learning and remembrance consolidation (Ramos-Rodriguez et al., 2013). In the current study, only the animals exposed to MSG-HD displayed a significant decline in the levels of the neurotransmitter (Dopamine, serotonin, epinephrine, and norepinephrine) of the brain tissue. This observation showed that the MSG-induced toxicity through the impairment of the neurotransmitters as Hashem, El-Din Safwat & Algaidi (2012) found that MSG changed the neurons and astrocytes in the cerebellar cortex of albino rats. Therefore, the present data indicated that the neurotransmitters are involved in protecting the brain from MSG-HD toxicity.

The catecholamine level was diminished as the dosage of MSG was elevated. Kardeşler & Başkale (2017) reported that the reduction in the catecholamine level empathized to the increment the MSG dosage that is a neurochemical reasoning for the learning disorderliness. Also, Tao et al. (2014) found that MSG injections caused changes in the brain’s hippocampus part which may result in a few behavioral alternations in the animals.

Madrigal et al. (2003) reported that COX-2 has some important brain functions as a synaptic activity, remembrance consolidation, functional hyperemia, and neuro-inflammation. The COX-2 level was increased significantly as the effect of MSG-HD that was a confirmation of a candid importance of COX-2 in neurodegenerative functions. The elevation of COX-2 in MSG-HD animals was confirmed by the histopathological changes as neurons are especially vulnerable to harm caused by free radicals creation through COX-2 peroxidase activity, whereas glial cells are more defiant (Madrigal et al., 2003). They also found that inhibition of COX-2 decreasing the membrane lipid peroxidation mediators and the decline of GSH strongly suggested that the COX isoform is elaborated in the aggregation of oxidative mediators erect in the brain.

Over-production of COX-2 has been correlated with neurotoxicity in hypoxia/ischemia and seizures (Donovan et al., 2000). Increasing the activity of COX-2 led to the formation of PGE2 that could be participated to synaptic malleability over different mechanisms, including inflection of adrenergic, noradrenergic, and glutamatergic neurotransmission, reshaping the actin in the cytoskeleton thus changing the shape of spines and dendrites, and controls the membrane excitability (Bazan, 2003). Considerable mechanisms could be provoked by COX-2 overproduction that could commit to oxidative stress-mediated harm by generating ROS during the peroxidase activity. Moreover, PGE2 has reinforced the effect of glutamate release. It could lead to the producing of more reactive free radical such as peroxynitrite. COX-2 expression within the brain has been related with pro-inflammatory activities, thought to be instrumental in neurodegenerative forms.

ZnO NPs inhibited the activity of COX-2 as compared to the control that could have beneficial to decline the inflammation and degenerative neuropathology.

There were not any histopathological changes in the ZnO NPs treatment-rats as comparing with normal animals. These results are consistent with Wang et al. (2008) and Zheng, Li & Wang (2009), who confirmed that ZnO NPs do not have any toxic effects on the brain tissues. In the contrasting context, Win-Shwe & Fujimaki (2011) reported that the toxic effects of ZnO NP on the CNS include cytotoxicity, inflammation, and oxidative stress induction that result in neurodegeneration. Gantedi & Anreddy (2012) found that 5 mg/kg ZnO NPs did not produce any significant alternations in histology of the brain of rats after a month of the treatment.

However, Elshama, El-Kenawy & Osman (2017) found that ZnO NPs led to the degeneration of the brain and spinal cord including pyramidal and glial cells, white and gray matter related with pyknotic nuclei and disturbance in many cytoplasmic organelles as well as changes in ultrastructural of the brain and spinal cord depending on its dose. The present study differed from the observations of Elshama, El-Kenawy & Osman (2017) in the effect of ZnO NPs could be due to the difference in the dose, route and the duration of exposure. Histopathological examination of rat brains exposed to ZnO NPs showed normal structure as the control animals.

There were no data available in the literature about the effect of ZnO NPs/GTE mixture on the brain. The mixture has increased the activities of SOD, CAT, GPx, and levels of GSH, and BDNF of the brain tissue than that the effect of ZnO NPs or GTE separately. Also, it has the ability to reduced MPO and XO than the ZnO NPs.

The action of Zn ONPs/GTE on MSG-induced brain toxicity can be due to its role on various neurotransmitters or due to its antioxidant property as the CAT, SOD, GPx and GSH enhanced as well as restored the other neural biomarkers parameters. This indicates that the mixture can decrease the harm caused by the induction of MSG-HD. This effect is attributable to the antioxidant character of the mixture which helps to scavenge free radicals thus prevent the tissues from damage.

Treating the rats with ZnO NPs/GTE mixture decreased the COX-2 activity in MSG-HD group that directly decreased the level of BDNF in the brain tissue.

The histopathological examination of the brain of rats treated with different dosages of MSG showed marked changes in the tissue with MSG-HD only. Severe damage occurred in the brain cell of rats that represented in focal gliosis, cellular edema, focal hemorrhage, and necrosis. Such changes also are in parallel with biomarkers changes represented in the activities of the brain antioxidant enzymes (SOD, CAT, GPx, MPO, XO, AChE, and COX-2) and levels of BDNF and PGE2. However, this improvement was not complete. A few pathological injuries still hold on but to less degree in MSG-HD with ZnO NPs/GTE.

Conclusions

In conclusion, MSG suspiciously affects the neurotransmitters of rats as well as to its effects on biomarkers of the brain parameters such as BDNF, COX-2 and PGE2 and antioxidant enzymes activities (SOD, CAT, and GPx). MSG caused neuronal degeneration in the cerebral cortex that supported those findings alternation in neurochemical biomarkers. The results of COX-2 revealed that it is one of the mechanisms through which stress may lead to cellular oxidative status in the brain by MSG-HD. Therefore, ZnO NPs/GTE proved to be benefit in decreasing the toxicity of MSG and ZnO NPs due to its conjugation with GTE. Its ameliorated action could be related to its ability to remove the free radical.

Supplemental Information

Supplemental Information 1 Physiological raw data

Click here for additional data file.

Additional Information and Declarations

Competing Interests

Author Contributions

Animal Ethics

Data Availability

The authors declare there are no competing interests.

Reham Z. Hamza conceived and designed the experiments, performed the experiments, analyzed the data, contributed reagents/materials/analysis tools, prepared figures and/or tables, authored or reviewed drafts of the paper, approved the final draft.

Fawziah A. Al-Salmi conceived and designed the experiments, performed the experiments, contributed reagents/materials/analysis tools, prepared figures and/or tables, authored or reviewed drafts of the paper.

Nahla S. El-Shenawy analyzed the data, prepared figures and/or tables.

The following information was supplied relating to ethical approvals (i.e., approving body and any reference numbers):

This experimental study was performed with the confirmation of the local ethics committee on use and care for animal experiments at Taif University of Biological Department (permit number: 39-31-0034).

The following information was supplied regarding data availability:

Raw data is available as a Supplemental File.

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
