# Peer review of "Evaluation of the effects of the green nanoparticles zinc oxide on monosodium glutamate-induced toxicity in the brain of rats"

_PeerJ, doi:10.7717/peerj.7460_

## Round 0.1 · original submission · Major Revisions

Please incorporate reviewer comments into your revision and/or address those comments as appropriate. One of the reviewers included an annotated manuscript.

Reviewer 1 ·

Basic reporting

The general outline is satisfactory although the manuscript would benefit from English language editing

Experimental design

1. Original primary research is within Aim and Scope of the journal
2. The Research question is not clearly defined and would require refining
3. The investigations were performed to a satisfactory technical and ethical standard
4. Methods were not described in sufficient detail to allow for replication

Validity of the findings

There is some novelty in the study although the authors have not clearly defined this.
All underlying data have been provide, the images do not represent region studied
The conclusions are well stated although the inferences drawn are not supported by the results

Additional comments

Topic: is it ……Zinc Oxide Green Nanoparticles….. or …..Zinc Oxide Green Tea Nanoparticles…..?
Abtract: alteration not alternation. Cortices not cortexes. Duration of administration of msg was not mentioned. methods and results are being mixed together. What is MSG-HD?
Introduction: Line 35 ‘because they are safe’ safe for what?
Line 35-39 should be revised for clarity of language
Line 46- is this work about evaluating the toxicity of ZnO NPs? if not, then why this line of thought?
Line 51-60- please revise for clarity
Line 68-70- needs English language editing
Line 79-84- the purpose of this paragraph is not clear
Line 85-92- How the contents relate to the present study is not clear
Line 93- This statement is UNTRUE

Animals and ethical considerations
Rats were chosen as the most appropriate animal species for the
experiment……….. Why?
The sample size was chosen based on the lowest number of animals to get actual
results………..What does this statement mean?
What is the basis for the doses of green tea and MSG used in this study?
Estimation of some bioactive compounds as markers for neuropathic pain….. This heading should be changed because what is being measured here is not neuropathic pain
Histopathological slides
There are inconsistencies in the areas of the brain from where the photomicrographs were taken with some reflecting the cortex (slides c, d and g) and a few reflecting the cerebellum (slide a) hippocampus (slide b) or deeper structures (slide f). Also, there is inconsistency of magnification. These problems make the claims regarding morphological changes unverifiable.



Discussion
Line 254….. Microglial actuation produces high amounts of free radicals that can harm proteins, lipids, and nucleic acids at the site of particle deposition ……Please explain the appropriateness of this sentence for this section.


Line 277….. Dopamine and catecholamine are neurotransmitters that occur abundantly in the CNS…….Do the authors consider catecholamine a distinct neurotransmitter?

Reviewer 2 ·

Basic reporting

It's good to publish.

Experimental design

It's good to publish.

Validity of the findings

It's good to publish.

Additional comments

It's good to publish.

Reviewer 3 ·

Basic reporting

- This article is well written. The level of English is of high quality. Nevertheless, there are some few grammatical and typing mistakes. see lines 21;38
- The literature cited in this work are relevant to the study and the topic.
The authors should improve in the way some references are written in the references to avoid disparities. See lines 399;406;408 and 430
- The authors provided raw data of the study in a good format and the figures and tables of the work are of high quality. The figures and tables depict resulst that are relevant to the hypotheses.

Experimental design

- The work present is an original research article within the aims and scope of Peer J.
- This research is innovative it gives supplementary information about the pathways involved in monosodium glutamate toxicity. The authors should improve the problem statement. See lines 80-44 and line 93.
- The methodology use in this study is appropriate, an ethical clearance for the use of the animal in laboratory studies was obtained. The author describes the protocol the used in a manner that can be replicable.

Validity of the findings

- The author feet this work in the previous works of their laboratory team and also they added additional information about the toxicity of MSG and the possible beneficial effect of natural products against MSG induced neurotoxicity and brain damages.
- The statistical analysis done hear is appropriate to this type of comparative effect of natural /synthetics products in vivo.

Additional comments

The work presented in this article is of great interest and give additional information about MSG mediated neurotoxicity. Nevertheless, we suggest that the authors your proofread the manuscript in order to reduce the levels of abbreviations, grammatical/typing errors. In the introduction the authors should also clearly state the rationale of the study.

Annotated reviews are not available for download in order to protect the identity of reviewers who chose to remain anonymous.

---

## Round 0.2 · accepted · Accept

Thank you for your efforts in revising your manuscript.

Reviewer 1 ·

Basic reporting

I am now satisfied.

Experimental design

Required corrections are now done

Validity of the findings

Satisfactory

Additional comments

None

Reviewer 3 ·

Basic reporting

The article has benefited from English language editing and the final document is well written.
The literature cited in this work are relevant to the study and raw data of the study is in an appropriate format.

Experimental design

no comment

Validity of the findings

no comment